# Brain Hsp90 Inhibition Mitigates Facial Allodynia in a Rat Model of CSD Headache and Upregulates Endocannabinoid Signaling in the PAG

**DOI:** 10.3390/ph18101430

**Published:** 2025-09-24

**Authors:** Seph M. Palomino, Aidan A. Levine, Erika Liktor-Busa, Parthasaradhireddy Tanguturi, John M. Streicher, Tally M. Largent-Milnes

**Affiliations:** Department of Pharmacology, University of Arizona, 1501 N. Campbell Avenue Tucson, Tucson, AZ 85719, USA; seph.palomino@utsouthwestern.edu (S.M.P.); htc9007@nyp.org (A.A.L.); erikal@arizona.edu (E.L.-B.); parthasaradhit@arizona.edu (P.T.); jstreicher@arizona.edu (J.M.S.)

**Keywords:** cortical spreading depression (CSD), migraine, heat-shock protein 90 (Hsp90), 17-AAG, anandamide (AEA), N-acyl phosphatidylethanolamine phospholipase D (NAPE-PLD), fatty acid amide hydrolase (FAAH), cannabinoid

## Abstract

**Background/Objectives**: The role of the molecular chaperone heat shock protein 90 (Hsp90) in pain and analgesia has been recognized; however, no study to date has investigated its role in facial allodynia during headache. In the current study, we examined the role of Hsp90 and its possible connection to the endocannabinoid system utilizing a rodent model of cortical spreading depression (CSD). **Methods**: CSD, a physiological phenomenon associated with headache disorders, was induced by cortical injection of KCl in female Sprague Dawley rats. To selectively inhibit Hsp90, 17-AAG was applied on the dura mater 24 h before CSD induction. Periorbital allodynia was assessed by von Frey filaments, while tissue samples were subjected to LC-MS, qPCR, Western immunoblotting, and the GTPγS coupling assay. **Results**: Increased expression of Hsp90 was selectively observed in the periaqueductal gray (PAG) harvested 90 min after cortical KCl injection, suggesting increased cellular stress from CSD induction. Application of 17-AAG (0.5 nmol) on dura mater 24 h before CSD induction significantly prevented facial allodynia as measured by von Frey filaments. This effect was blocked by injection of the CB_1_R antagonist rimonabant (1 mg/kg, ip). The pretreatment with 17-AAG significantly increased the level of anandamide (AEA) in PAG 90 min after cortical insult, as measured by LC-MS. This effect was accompanied by reduced expression of FAAH and increased expression of NAPE-PLD in the same nuclei. **Conclusions**: These results suggest that Hsp90 inhibition positively modulates the endocannabinoid system, causing pain relief through descending pain modulation in PAG post-CSD.

## 1. Introduction

Recent epidemiology surveys revealed that nearly 100 million Americans suffer from some sort of chronic pain; among them, more than 30 million patients have migraines [1,2]. Migraine is a complex neurological disorder characterized by severe headaches and other accompanying symptoms which can seriously impair patients’ quality of life. Cortical spreading depression (CSD) is considered an electrophysiological correlate of migraine aura, but there is a known link between CSD and other neurological disorders, like stroke and traumatic brain injury (TBI) [3]. CSD is characterized by a massive increase in extracellular K^+^ and glutamate, as well as ionic and pH shifts [4]. CSD is also able to activate central and peripheral nociceptive pathways, as we previously demonstrated in agreement with former literature data [5,6]. Despite the high prevalence and severity of migraine symptoms, including aura, available therapies (i.e., triptans) reduce pain intensity and duration in only about 30% of patients [7]. This limitation for the treatment of migraine pain emphasizes the need to discover and validate novel, more effective targets.

Heat shock protein 90 (Hsp90) is a ubiquitous and highly expressed molecular chaperone, responsible for the regulation of protein folding of specific client proteins like transcription factors, nuclear hormone receptors, and signaling kinases [8]. Hsp90 plays an essential role in many cellular processes, including cell cycle control, cell survival, hormones and other signaling pathways [9]. The recognition of Hsp90 as a promoter for uncontrolled cell replication has led to the investigation of Hsp90 inhibitors as potential anti-cancer therapies; numerous second- and third-generation Hsp90 inhibitors are currently involved in clinical trials for the treatment of cancer [10,11]. Importantly, overexpression of Hsp90 has been linked to many other pathological conditions beyond cancer [7]. Elevated levels of Hsp90 were observed during viral infection, inflammation, and certain neurodegenerative diseases [12].

Novel results suggest that Hsp90 is a central regulator of acute signal transduction, which may link Hsp90 to the above disease states. It can directly regulate numerous G protein regulatory kinases (GRKs), as well as G protein coupled receptors (GPCRs). Moreover, Hsp90 can influence the localization of these proteins within different cell compartments. For instance, Hsp90 was shown to direct mature Gα subunits of GPCRs specifically to lipid rafts and mitochondria [13]. He et al. confirmed that Hsp90 has an impact on cannabinoid signaling by regulating the translocation of Gα_i2_ protein of cannabinoid type-2 receptor [14]. Hsp90 has been shown to form a complex with small GTPases, like Raf kinase, but it is also associated with members of the Rab, Rho, and Rac kinase families [15]. It is also known that Hsp90 influences the ERK-MAPK pathway signaling in the brain, but represses it in the spinal cord, influencing opioid signaling and opioid-induced analgesia [16,17]. These complex findings support that Hsp90 possesses a diverse range of cellular functions far beyond its originally identified role in protein folding and chaperoning. One of the recently recognized roles of Hsp90 is its involvement in pain signaling and analgesia [18].

Hsp90 has been shown by several research groups to regulate the initiation and perpetuation of different types of pain [18]. Hsp90 inhibitors significantly reduced hyperalgesia and allodynia caused by inflammatory-related pain models, such as peripheral neuropathy [19,20]. The administration of an Hsp90 inhibitor also mitigated the mechanical allodynia associated with monoarthritis in rats [21]. The available results suggest that the pharmacological blockade of Hsp90 possesses analgesic effects either indirectly by suppression of inflammatory cascades or more directly by neuroprotection and the promotion of antinociceptive signaling [18]. These findings support the therapeutic potential of Hsp90 inhibition in the treatment of pain. In our former work, we have demonstrated that the inhibition of Hsp90 can preserve blood–brain barrier integrity after CSD induction [22]. To date, no study has investigated the possible analgesic effect of Hsp90 inhibitors in any headache model; therefore, this study addressed this gap in knowledge by investigating how Hsp90 inhibition might affect headache-like behaviors induced by cortical injection of KCl (1M) to model CSD and the potential interactions of Hsp90 with the endocannabinoid system.

## 2. Results

### 2.1. The Expression of Hsp90 Increased in Female Rat PAG at 90 Minutes After Cortical KCl, but Not aCSF, Injection

Literature reports have shown that the expression of Hsp90 is enhanced after certain insults, including pain [8]. Therefore, we first investigated whether cortical injection of KCl, a model of CSD induction, or aCSF control altered total Hsp90 expression using Western blot analysis. Three brain regions of female rats (V1M cortex, PAG, and trigeminal ganglion [TG]), all regions associated with CSD (V1M) or pain (PAG, TG), were harvested 30 and 90 min after dural injection of 1 M KCl or aCSF control (Figure 1A). Interestingly, the peak of facial allodynia in the KCl-induced CSD model was observed at these 30 and 90 min time points in previous studies [6,23]. The expression of Hsp90 was not significantly changed by KCl injection in the V1M cortex (CSD induction point) or TG (dural activation by injection) samples at either time points assessed (Figure 1B,D) (aCSF vs. KCl: cortex 30 min: *p* = 0.6746, cortex 90 min: *p* = 0.3131, F(1,10) = 0.2611; TG 30 min: *p* = 0.6292, TG 90 min: *p* = 0.4787, F(1,12) = 1.056 by two-way ANOVA (CNS region × time), n = 3–4 in each group). In contrast, the expression of Hsp90 in PAG was statistically higher 90 min post KCl injection as compared to that in the aCSF control (Figure 1C) (aCSF vs. KCl: PAG 30 min: *p* = 0.7204, PAG 90 min: *p* = 0.0053, as assessed by two-way ANOVA, F(3,22) = 6.655, n = 4 in each group). These data indicate that CSD induction by cortical injection of KCl selectively increases the expression of Hsp90 within the PAG; this increase could then correlate with changes in pain or analgesia during CSD.

### 2.2. Periorbital Allodynia Associated with Cortical Spreading Depression Was Mitigated by Hsp90 Inhibition with 17-AAG

In the next set of experiments, we explored the role of Hsp90 inhibition in CSD-associated periorbital allodynia using pharmacological manipulation of Hsp90 by injection of 17-AAG (Figure 2A). Notably, the early clinical trials using Hsp90 inhibitors reported headache as a side effect, but a recent meta-analysis of the available clinical data (doses between 0.8 mg/kg and 53 mg/kg applied systemically) did not find a statistically significant increase in headache incidents after administration of Hsp90 inhibitors [24]. To fully execute the effect of Hsp90 inhibition on headache pain and considering that 17-AAG might induce headache-like behavior in rodents, we first tested the effect of Hsp90 blockade after a single application of 17-AAG. Using our former data for dosing, 0.5 nmol and 5 nmol doses in 5 uL of 17-AAG were applied to the dura, with the 5 nmol dose representing a suprathreshold dose [16,17]. Facial allodynia was assessed by application of von Frey filaments up to 4 h after injection. Acute administration of 17-AAG at either dose did not induce periorbital allodynia compared to vehicle groups (Figure 2B) (17-AAG (0.5 nmol) or 17-AAG (5 nmol) vs. control *p* > 0.05 up to four hours after dural application, as assessed by two-way ANOVA, F(12,176) = 1.671, n = 8–12/group).

In the next step, we combined the 17-AAG treatment with the aCSF control. The animals were observed 24 h post application of 17-AAG or vehicle before aCSF cortical injection, and no periorbital allodynia was observed (Figure 2C, baseline). After aCSF injection, animals receiving the suprathreshold dose of 17-AAG (5 nmol) developed significant periorbital allodynia, whereas the 0.5 nmol of 17-AAG did not cause significant change in facial withdrawal thresholds (Figure 2C) (17-AAG (5 nmol) + aCSF vs. vehicle + aCSF: 60 min *p* = 0.0089, 90 min *p* = 0.0036, 120 min *p* = 0.0048, 180 min *p* = 0.0003, 300 min *p* = 0.0005, as assessed by two-way ANOVA, F(14,214) = 1.776, n = 8–12/group). These data suggest that high doses of Hsp90 inhibition on the dura mater 24 h before insult increase susceptibility to develop tactile allodynia and further justify the selection of the 0.5 nmol dose for the rest of this study.

The subsequent experiments evaluating the effect of Hsp90 inhibition in combination with CSD used the highest dose of 17-AAG (0.5 nmol) that did not facilitate headache-like behaviors. Female rats were pretreated with either vehicle or 17-AAG (0.5 nmol) applied to the dura mater followed by CSD induction with 0.5 μL of 1 M KCl 24 h later. In control rats (vehicle + KCl), peak periorbital allodynia was observed at 90 min after cortical injection and lasted for 3 h (Figure 2D). Pretreatment with 17-AAG (0.5 nmol) significantly attenuated CSD-associated periorbital allodynia 90 min after KCl injection (Figure 2D) (17-AAG + KCl vs. vehicle + KCl at 90 min *p* = 0.0016, as assessed by two-way ANOVA, F(7, 159) = 1.722, n = 11–12/group). The corresponding AUC data (Figure 2E) supported the inhibitory effect of 17-AAG on periorbital allodynia caused by CSD (17-AAG + KCl vs. vehicle + KCl *p* = 0.0318, vehicle + KCl vs. vehicle + aCSF *p* < 0.0001, as assessed by one-way ANOVA, F(5,51) = 12.57, n = 11–12/group). Together, these data suggest that Hsp90 plays a role in cortical KCl-induced periorbital allodynia and pretreatment with the Hsp90 inhibitor 17-AAG can reduce the magnitude of this headache-like behavior.

### 2.3. The Effects of Hsp90 Inhibitor on Periorbital Allodynia Was Reversed by the CB_1_R Antagonist Rimonabant

In the next set of experiments, we tested how the blockade of the cannabinoid receptor CB_1_R can influence the preventative effects of 17-AAG in mitigating periorbital allodynia. The animals were pretreated with 17-AAG 24 h before CSD induction (as described above), followed by the CB_1_R inverse agonist/antagonist, rimonabant (SR141716A, 1 mg/kg, i.p.) 30 min before the cortical injection of KCl (Figure 2F). Dosing rimonabant before CSD induction mitigated the anti-allodynic effect of 17-AAG at the 90 min time point (Figure 2F,G) (Figure 2F: vehicle + KCl vs. 17-AAG + KCl *p* = 0.0099, vehicle + KCl vs. 17-AAG + KCl + rimonabant *p* = 0.1296 as assessed by two-way ANOVA, F(3.767,100.4) = 41.08, n = 9–12/group) (Figure 2G: AUC for vehicle + KCl vs. 17-AAG + KCl *p* < 0.0001, vehicle + KCl vs. 17-AAG + KCl + rimonabant *p* = 0.2023, 17-AAG + KCl vs. 17-AAG + KCl + rimonabant *p* = 0.0003, as assessed by one-way ANOVA, F(2,27) = 24.78, n = 9–12/group).

Of the animals pretreated with 17-AAG, 4/10 (40%) were classified as sensitive to periorbital probing, which was defined as having periorbital withdrawal thresholds < 6 g at two or more consecutive time points within 120 min of the KCl injection (CSD induction); this was lower than in the vehicle-pretreated group, 11/12 (91.6%) (Figure 2H). Administration of rimonabant increased the number of animals showing tactile sensitivity to 8/10 (80%) from the 4/10 (40%) observed in the absence of cannabinoid antagonist (Figure 2H). These data suggest that the pain-relieving effect of Hsp90 inhibition during CSD is CB_1_R-dependent despite the lack of protein expression or signaling changes with Hsp90 inhibition above.

### 2.4. Inhibition of Hsp90 Did Not Change the Expression Levels of CB_1_R and AHA1 Protein in the PAG After CSD Induction

Hsp90 acts as a chaperone protein and previous studies have shown overexpression of AHA1, one of the co-chaperones of Hsp90, enhanced the surface levels of the CB_1_R in HEK cells; however, Hsp90 inhibitors did not affect the total expression levels of AHA1 in vitro [25]. Thus, the next studies tested the effect of Hsp90 inhibition on cannabinoid receptor expression as well as AHA1 expression in PAG after CSD induction. PAG samples 90 min post-KCl/aCSF injection were used since increased Hsp90 expression after KCl injection was observed at that time point and location. PAG tissue from female rats pretreated with vehicle or 0.5 nmol of 17-AAG (same dosage was used in the work of Lei et al., 2017 [16]) 24 h before dural injection of KCl or aCSF were processed and analyzed (Figure 3A). KCl and veh + KCl groups were combined as no statistical difference was observed in these controls. Real-time qPCR results showed that levels of *Cnr1* (CB_1_R) mRNA were significantly different between the vehicle + KCl- vs. 17-AAG + KCl-treated samples (Figure 3B) (vehicle + KCl vs. 17-AAG + KCl *p* = 0.0500, as assessed by one-way ANOVA, F(2,21) = 1.845, n = 6–16/group). The *Cnr2* mRNA level in PAG was significantly reduced after KCl injection as compared to the aCSF control, but pretreatment with 17-AAG did not influence the *Cnr2* mRNA level as compared to KCl (Figure 3C) (aCSF vs. KCl *p* = 0.0110, as assessed by one-way ANOVA, F(2,23) = 5.464, n = 8–12/group).

The next studies focused on the total protein expression of CB_1_R and AHA1 (as justified above). CB_2_R protein was not evaluated given the lack of changes after Hsp90 inhibition and the paucity of reliable antibodies. PAG samples harvested as described above were subjected to Western immunoblotting. No significant difference in the total expression of CB_1_R and AHA1 was observed between treatment groups (Figure 3D,E). (CB_1_R expression: aCSF vs. KCl + vehicle *p* = 0.1507, KCl + vehicle vs. KCl + 17-AAG *p* = 0.8468 F(2,12) = 2.702, as assessed by one-way ANOVA n = 4/each group; AHA1 expression: aCSF vs. vehicle + KCl *p* = 0.7169, vehicle + KCl vs. 17-AAG + KCl *p* = 0.5614, F(2,13) = 0.6568 as assessed by one-way ANOVA, n = 4 in each group). These results indicate that the blockade of Hsp90 does not influence the total protein levels of CB_1_R and AHA1 at the 90 min time point after cortical insult; however, the reduced mRNA level of CB_1_R in the 17-AAG-pretreated group raises the possibility that Hsp90 inhibition could still impact the activity or function of CB_1_R.

### 2.5. CB_1_R Function in PAG Is Compromised After Cortical KCl Injection

To determine if the function of the CB_1_R in the PAG was changed by Hsp90 inhibition, a ^35^S-GTPγS binding assay was performed using the CB_1_R-selective agonist PrNMI [26]. Female rats were pretreated with 17-AAG or vehicle 24 h before KCl injection, as described above. PAG samples were harvested 90 min post KCl injection. Tissue was also harvested from aCSF control as well as KCl-injected rats (without vehicle injection) 90 min post cortical injection as controls (Figure 4A). PrNMI returned agonist curves for each group; visually, the KCl treatment caused an apparent loss of agonist efficacy (E_MAX_) that was not altered by the 17-AAG treatment (Figure 4B). These observations were quantified by analysis of the potency (EC_50_) and efficacy in each group (Figure 4C). The KCl treatment significantly reduced the E_MAX_ for all groups, which was not altered by the 17-AAG treatment. In contrast, the EC_50_ was not significantly altered between groups. These results suggest partial desensitization of the CB_1_R in the PAG under CSD conditions. These results cannot be explained by changes in receptor expression since protein levels were quantified above and did not change, nor by receptor availability as the assay homogenizes the membrane compartment of the tissue. However, 17-AAG treatment did not alter this effect, excluding changes in unit receptor signaling as a mechanism for the effects of Hsp90 inhibition.

### 2.6. Hsp90 Inhibition Increases AEA Levels in PAG After CSD Induction

Our former results showed that dural injection of KCl significantly and selectively reduced 2-AG levels within the PAG but not the V1M cortex, TG, or trigeminal nucleus caudalis (Vc), while the AEA levels were unchanged in any region [27]. The next experiment examined whether the levels of endogenous cannabinoid lipids 2-arachidonoylglycerol (2-AG) or anandamide (AEA) in PAG were altered by Hsp90 inhibition in the CSD model (Figure 5A). Animals were treated with 17-AAG (0.5 nmol) or vehicle 24 h before CSD induction. PAG samples were harvested at 90 min after cortical KCl injection and subjected to LC-MS. In addition, 24 h pretreatment with 0.5 nmol 17-AAG applied onto the dura mater caused significant elevation in AEA levels in PAG compared to vehicle + KCl control (Figure 5B; AEA: vehicle + KCl vs. 17-AAG + KCl *p* = 0.0052, t(8) = 3.802, as assessed by unpaired *t*-test, n = 5 in each group), but did not change the 2-AG levels (Figure 5C; 2-AG vehicle + KCl vs. 17-AAG + KCl *p* = 0.1845, t(8) = 1.434, as assessed by unpaired *t*-test, n = 5 in each group). These results suggest a mechanism by which Hsp90 inhibition can boost CB_1_R signaling with AEA to reduce headache pain without altering protein expression or intrinsic signaling, as shown above.

The following experiments were designed to understand how AEA levels were elevated after application of 17-AAG (Figure 5A). The level of AEA is primarily controlled by the synthetic enzyme NAPE-PLD and the degradative enzyme FAAH in the PAG [28,29]. 17-AAG application is reported to upregulate heat shock protein 70 (Hsp70), which can increase intracellular lipid trafficking, including AEA [16,30]. Therefore, PAG samples were harvested at the 90 min time point from female rats pretreated with 17-AAG (0.5 nmol) or vehicle followed by cortical injection of KCl and subjected to Western immunoblotting targeting NAPE-PLD, FAAH and Hsp70 proteins (Figure 5D–F). NAPE-PLD levels were statistically elevated within PAG harvested from rats pretreated with 17-AAG, suggesting the increased synthesis of AEA (Figure 5D) (vehicle + KCl vs. 17-AAG + KCl *p* = 0.0208, t(6) = 3.110, as assessed by unpaired *t*-test, n = 4 in each group). Compared to the vehicle control, pretreatment with 17-AAG significantly reduced the FAAH levels in PAG, suggesting reduced degradation of AEA (Figure 5E) (vehicle + KCl vs. 17-AAG + KCl *p* = 0.0020, t(10) = 4.157, as assessed by unpaired *t*-test, n = 4–8/group). No difference was observed in protein levels of Hsp70 between vehicle and 17-AAG groups in PAG tissue (Figure 5F) (vehicle + KCl vs. 17-AAG + KCl *p* = 0.3746, t(10)= 0.9294, as assessed by unpaired *t*-test, n = 4–8/group). Together, these data suggest that 17-AAG increases AEA levels by increasing NAPE-PLD and decreasing FAAH expression but not through changes in total Hsp70 expression.

## 3. Discussion

Emerging reports confirm the role of Hsp90 in pain signaling and analgesia, but our pioneering work describes the contribution of Hsp90 in migraine-like pain after CSD. Our former work indicated the effect of Hsp90 inhibition 24 h after its administration in mice models [17]. Therefore, 17-AAG was administered in the same setting in the current CSD model and showed anti-allodynic effect in the KCl-induced CSD in female Sprague Dawley rats. This effect was mitigated by the CB_1_R antagonist rimonabant and was accompanied by increased expression levels of NAPE-PLD, responsible for AEA synthesis, and reduced expression of the degradative enzyme FAAH in the periaqueductal grey (PAG), without influencing the intrinsic signaling capacity of the CB_1_R confirmed by GTPγS binding assay. These later observations also support the 24 h pretreatment schema of 17-AAG. This time frame could possibly allow 17-AAG to modulate the activity of Hsp90-related proteins, including transcription factors, leading to the modification in total protein expression, manifested as changes in the total level of AEA metabolic enzymes. Together, these results explain our observed increase in AEA levels in the PAG with Hsp90 inhibitor treatment. These results suggest that Hsp90 promotes pain transmission during the CSD event in the PAG, perhaps by repressing the levels of AEA, and therefore can be a new, potential target in the treatment of migraine-like headache pain.

Our study mainly focused on one brain area, the periaqueductal grey (PAG). PAG is one of the most important brain centers, responsible for the ascending and descending modulation of pain perception, and it has a special role in the induction and maintenance of migraine-like pain [31]. The activation of PAG during and after migraine is well documented [32]. The PAG region can remain active post-cessation of pain by treatment with triptans [33], implicating it as an important region modulating nociception from the trigeminovascular system. Our former results using the rodent model of CSD also confirm the importance of PAG area in the development of cortical spreading depression [27]. It is well known that Hsp90 can be upregulated in several disease states in various central and peripheral tissues [8]. However, we found elevated Hsp90 levels specifically in the PAG within the examined pain centers after induction of CSD, which could indicate increased cellular stress from the noxious stimulus. Considering the impact of CSD leading to the onset of neurogenic inflammation, the result of Hsp90 protein levels not being elevated in the other regions associated with ascending pain transmission was not anticipated and could suggest a specific role for Hsp90 in the PAG post-CSD. Knowing that PAG is a primary site of central sensitization in CSD model, our data raise the possibility that Hsp90 might be one of the key molecular mediators of this process. Since elevated expression of Hsp90 was observed exclusively in PAG samples, the rest of our study focused on that specific pain-related nucleus.

Hsp90 inhibition has proven useful in pain research, specifically in inflammatory pain [19,20]. More recently, the pharmacological blockade of Hsp90 has also been shown to enhance the antinociceptive properties of morphine in post-surgical and neuropathic pain models through opioid receptor signal modulation [18]. The isoform-selective inhibitors of Hsp90 have been reported to improve the opioid therapeutic index through engagement of spinal circuits [34]. These results indicate that Hsp90 may have a role in distinct nociceptive signaling processes. Our data support that administration of the Hsp90 inhibitor 17-AAG can lessen the facial sensitivity caused by CSD and it can also reduce the number of allodynic animals. Since the literature suggests a link between Hsp chaperones and the endocannabinoid system, we tested if the anti-allodynic effect of 17-AAG is CB receptor-dependent. The inhibition of CB_1_R with rimonabant before CSD induction reduced the effect of 17-AAG, indicating the involvement of CB_1_ receptor in the analgesic activity of Hsp90 blockade.

These behavioral observations encouraged further investigation of the interaction between Hsp90 and the endocannabinoid system. 17-AAG pretreatment did significantly change the *Cnr1* mRNA expression in PAG compared to the non-treated CSD group. However, total protein levels of CB_1_R between those groups were unchanged. Moreover, there was no significant difference in the expression of AHA1, a co-chaperone of Hsp90 which has been reported to directly influence CB_1_R expression in vitro [25]. This result raised the possibility that Hsp90 inhibition might affect the functionality of CB_1_R, rather than absolute amount. Using a ^35^S-GTPγS functional assay, we observed the efficacy of PrNMI was decreased after CSD induction. However, this effect was not altered by Hsp90 inhibitor treatment. Taken together, these results suggest that Hsp90 inhibition reduces allodynia in the CSD model in a CB_1_R-dependent way, without alterations in CB_1_R expression or activity.

To further explore the antinociceptive mechanism of Hsp90 inhibition, the level of major endocannabinoids in PAG was measured by LC-MS. The LC-MS data revealed an elevated level of AEA in PAG after 17-AAG treatment, suggesting that endocannabinoid lipids are directly impacted by Hsp90 inhibition. The level of AEA is tightly regulated by the enzymes NAPE-PLD and FAAH in the central nervous system [29]. Our Western results show that Hsp90 inhibition increased NAPE-PLD expression while reducing FAAH levels in the PAG in post-CSD rats, thus providing a possible mechanism for elevated AEA. How 17-AAG inhibition elicits these changes in expression requires more investigation; however, the observed changes in the total level of AEA enzymes suggest that Hsp90 inhibition can possibly influence the transcription/translation machinery. During the 24 h pretreatment timeline, 17-AAG could modulate Hsp90-related transcription factors and consequently alter the protein expression. Some research points to Hsp90 as playing a role in regulating mRNA levels and protein half-life of the transcription factor HNF4A. This transcription factor promotes NAPE-PLD expression [35]; Hsp90 chaperone machinery database: https://www.picard.ch/Hsp90Int/index.php (accessed on 6 November 2023). Moreover, 17-AAG has been reported to downregulate STAT3, a promoter of FAAH expression [36,37]. Interestingly, cycloheximide, an inhibitor of protein translation abolished the additive effect of 17-AAG on the morphine-induced antinociception in mice, implying the role of protein translation in the effect of Hsp90 inhibition [17]. Future studies may build on these reports to elucidate the exact Hsp90/eCB enzyme interaction.

Hsp70 is a known intracellular chaperone of AEA in the cytosol [30]. Some papers have also reported increased Hsp70 levels after inhibition of Hsp90 by 17-AAG administration [16]. Therefore, expression of Hsp70 was also measured by Western immunoblotting after 17-AAG treatment. We found no difference in levels of Hsp70 between the 17-AAG-treated and the vehicle group in PAG tissues 90 min post KCl induction, which may reflect different routes of administration, species differences, or sex differences. Overall, we can conclude that 17-AAG inhibition of Hsp90 increases AEA in the PAG tissue via increased synthesis and decreased hydrolysis of AEA and not by changing transport via Hsp70; data cannot rule out actions through Hsp90/70 interactions.

It is well known that AEA possesses analgesic effects through the activation of CB_1_R [38]. Interestingly, Lau and co-workers showed that exogenous application of AEA, but not 2-AG, produced a reduction in inhibitory GABAergic transmission in PAG neurons, and this disinhibition can lead to the analgesic effect of AEA [28]. These former data align well with our results. The increased AEA level in PAG after 17-AAG administration can contribute to the anti-allodynic effect of Hsp90 blockade; however, its impact on GABA transmission needs to be investigated in future studies.

In addition to cannabinoid receptor activity, AEA has been reported to play a role in additional phenomena associated with migraine. A study published by Akerman and co-workers shows that AEA inhibited CGRP-induced and NO-induced dural vessel dilation, the effect of which was reversed by CB_1_R antagonist AM251 [39]. It was also reported that AEA induced a significant decrease in the nociceptive behavior in nitroglycerine-induced migraine model in male Sprague Dawley rats by reducing neuronal activation in nucleus trigeminal caudalis [40]. Using tract tracing and gene-editing techniques may help answer the question of what circuits are most impacted by Hsp90 inhibition in the PAG. Stimulation of the TRPV1 receptor by endocannabinoids in the PAG can also lead to analgesic effects [41]. Given the local increase in AEA within the PAG and our rimonabant results, this mechanism is unlikely but cannot be fully eliminated.

Review papers have emphasized the importance of peripheral FAAH inhibition as a possible therapeutic avenue for migraine pain [42,43]. Recent work has noted the importance of the central endocannabinoid system, especially 2-AG, in PAG during headache [44]. It is also known that increased functional activity of PAG was observed during migraine pain, which can play a role in central sensitization [32]. This work now demonstrates a significant contribution of the central AEA-endocannabinoid system in PAG during migraine-like pain.

Targeting FAAH-regulated AEA signaling has potential in the development of new agents against migraine pain. The results published by Cupini et al. showed increased FAAH activity and reduced AEA levels in platelets obtained from migraineur women [45]. The extensive research for compounds that can modulate endocannabinoid tone resulted in several inhibitors targeting FAAH. The global FAAH inhibitor, URB597, as well as the peripherally restricted FAAH inhibitor, URB937, significantly inhibited nociceptive processing in nitroglycerine-induced migraine models [43]. Recent safety concerns about individual FAAH inhibitors have delayed the progress of that research [46], though clinical trials are on-going with different individual agents (clinicaltrials.gov)

Our study is the first to describe a direct region-specific modulation of the FAAH-NAPE-PLD-AEA system by an Hsp90 inhibitor, further contributing to a mechanistic understanding of CSD-associated headache disorders and the importance of CB_1_R activity in the PAG for mitigating headache-associated allodynia. Hsp90’s ubiquitous expression and broad network of client protein interactions complicates its candidacy as a selective drug target. Although early generation Hsp90 inhibitors failed in clinical trial settings due to hepatotoxicity, this adverse effect has not been observed with more recent compounds that selectively target the C-terminal domain or disrupt co-chaperone interactions [18]. Overall, Hsp90 inhibitor, 17-AAG has displayed great utility in the preclinical setting both as a pharmacological probe and as a mechanistic tool for elucidating pathways involved in neuroinflammation and GPCR signaling. Given the specific activity observed in the PAG, this study further supports investigation of next-generation Hsp90 inhibitors in headache-associated pain models. Their potential to modulate other receptor systems implicated in headache pathophysiology may be leveraged for therapeutic development.

Limitations: The authors acknowledge existing limitations of this study. First, only female Sprague Dawley rats were tested in this paper since migraine affects females to males at a ratio of approximately 3:1. Further experiments need to be performed in the future to test our findings in male animals. Second, the Hsp90 inhibitor, 17-AAG, was administered 24 h before cortical injections of KCl or aCSF through the guide cannula onto dura mater. Future studies are therefore warranted to examine the effect of other routes and different timelines of Hsp90 inhibitor administration. Furthermore, this paper mainly focuses on the PAG brain area. There is a possibility that other brain regions are also involved in the interaction of Hsp90 with the endocannabinoid system. Lastly, all antibody-based assays can be biased by lot variability, changes in access to antigen binding, or non-specific binding; thus, the differences in antibody detection may reflect those changes, rather than modification in total protein expression.

## 4. Material and Methods

### 4.1. Drugs and Reagents

7-(Allylamino)-17-demethoxygeldanamycin (17-AAG) (#AAJ66960MC) was purchased from Alfa Aesar (Haverhill, MA, USA). Rimonabant (SR141716A) was purchased from Cayman Chemicals (Ann Arbor, MI, USA).

### 4.2. Animals

Intact, female Sprague Dawley rats (200–250 g, n = 175) were purchased from Envigo (Indianapolis, IN, USA) and housed in a climate-controlled room on a regular 12/12 h light/dark cycle with lights on at 7:00 am with food and water available ad libitum. Animals were initially housed 3 per cage but individually housed after the dural cannulation. All procedures were performed during the 12 h light cycle and according to the policies and recommendations of the International Association for the Study of Pain and the NIH guidelines for laboratory animals, and with IACUC approval from the University of Arizona (Approval Code#: 17-223, approval date: 2 September 2023). Justification for animal numbers was consistent with NIH policy (NOT-OD-15-102), and experiments were randomized to blinded treatment groups to give 80% power to detect a treatment effect size of 20% compared to a baseline response of 5% at a significance level of 0.05 [47]. Numbers required to achieve statistical power were determined by G.Power3.1 [44]. To minimize bias, animals were randomly assigned to treatment groups. Female rats were used as headache disorders affect females to males at a ratio of nearly 3:1 [48]. The stage of estrus cycle was not determined in this study.

### 4.3. Dural Cannulation

Dural cannulation was performed as previously described [6,23,49]. Briefly, anesthesia was induced with intraperitoneal 45:5:2 mg/kg cocktail of ketamine/xylazine/acepromazine. Rats were placed in a stereotactic frame (Stoelting Co., Wood Dale, IL, USA), and a 1.5 to 2 cm incision was made to expose the skull. A 0.66 to 1 mm hole (Pinprick/KCl: −6 mm A/P, −3 mm M/L from bregma) was made with a hand drill (DH-0 Pin Vise; Plastics One, Roanoke, VA, USA) to carefully expose the dura. A guide cannula (0.5 mm from top of skull, 22 GA, #C313G; Protech International Inc., Boerne, TX, USA) was inserted into the hole and sealed into place with glue. Two additional 1 mm holes were made caudal to the cannula to receive stainless-steel screws (#MPX-080-3F-1M; Small Parts, Miami Lakes, FL, USA), and dental acrylic was used to fix the cannula to the screws. A dummy cannula (#C313DC; Plastics One) was inserted to ensure patency of the guide cannula. Rats were housed individually and allowed 6–8 d to recover. Cannula placement and dural integrity at screw placement were confirmed postmortem; animals with incorrect placement (beyond V1M) or dural damage were excluded based on a priori criteria (no animals were excluded on these bases).

### 4.4. Cortical Injection of KCl or aCSF

Cortical injections were performed using a Hamilton injector (30 GA, 80308 701 SN, Hamilton Company, Reno, NV, USA) customized to project 1.0 mm beyond the dura into the occipital cortex [6]. The injector was inserted through the guide cannula to deliver a focal injection of 0.5 µL of 1 M KCl or artificial CSF (aCSF) into the cerebral cortex. aCSF comprised 145 mM NaCl, 2.7 mM KCl, 1 mM MgCl_2_, 1.2 mM CaCl_2_, and 2 mM Na_2_HPO_4_ (pH 7.4), and the solution was passed through a 0.2 µm syringe filter before injection. Cortical injections were always considered at t = 0 min.

### 4.5. Pre-Cortical Injection Treatments

A total of 5 uL of 17-AAG was applied onto the dura mater 24 h before cortical injections at concentrations of 0.05, 0.5, and 5 nmol utilizing the guide cannula in place for cortical injections. 17-AAG was dissolved in 100% DMSO to a concentration of 10 mM stock, and then diluted in saline for final concentration (0.05 and 0.5 nmol). The 5 nmol concentration dose was made from the same 10 mM stock into DMSO-Tween80-saline (1:1:80, *v*/*v*/*v*) due to solubility difficulties. Rimonabant (CB_1_R inverse agonist/antagonist, 1 mg/kg, 1 mL/kg) was injected intraperitoneally 30 min before cortical KCl injection.

### 4.6. Assessment of Periorbital Mechanical Allodynia

Periorbital allodynia was evaluated at baseline, after 17-AAG injection (acute response) and after cortical injection (t = 30, 60, 90, 120, 180, and 300 min) by an observer blinded to drug condition. Rats were grouped based on their post-surgical baseline to ensure equivalent pre-injection thresholds (6–8 g). Rats were acclimated to the testing box 1 h prior to evaluation of periorbital mechanical allodynia with calibrated von Frey filaments (1.0, 1.4, 2.0, 4.0, 6.0, 8.0 g) as previously described by Fioravanti and coworkers [49]. Behavioral responses were determined by applying calibrated von Frey filaments perpendicularly to the midline of the forehead at the level of the eyes with enough force to cause the filament to slightly bend while held for 5 sec while the animals were freely moving. A response was indicated by a sharp withdrawal of the head, vocalization, or severe batting at the filament with attempts to bite it. The withdrawal threshold was determined using a modified version of the Dixon up–down method. Briefly, the stimulus was increased one increment after a negative response, and it was decreased one increment after a positive response. The stimulus was progressively increased until a positive response was detected, and then decreased until a negative result was obtained. This “up-down” method was repeated until three changes in behavior were observed. Animals with a post-surgical baseline threshold of <80% maximum (6g) were considered sensitive according to inclusion/exclusion criteria and removed from behavioral analyses (n = 7).

### 4.7. Tissue Harvest

Rats were anesthetized with a ketamine/xylazine mix (80:10 mg/kg, i.p.), and then transcardially perfused with ice cold 0.1 M phosphate buffer at flow rates so not to burst microvasculature (i.e., 3.1 mL/min). After decapitation, periaqueductal grey (PAG) was dissected (from bregma AP: −6.5 mm to −8.3 mm, ML: ±0.5 mm, DV: −5.0 mm to −6.0 mm), flash-frozen in liquid nitrogen and stored at −80 °C until further use.

### 4.8. Tissue Preparation for Western Immunoblotting

On the day of preparation, samples were placed in ice-cold lysis buffer (20 mm Tris-HCl, 50 mM NaCl, 2 mM MgCl_2_ × 6H_2_O, 1% *v*/*v* NP40, 0.5% *v*/*v* sodium deoxycholate, 0.1% *v*/*v* SDS; pH 7.4) supplemented with protease and phosphatase inhibitor cocktail (Halt™ Protease and Phosphatase Inhibitor Cocktail, ThermoScientific, Waltham, MA, USA). All subsequent steps were performed on ice or at 4 °C. The samples were sonicated, and then centrifuged at 12,000× *g* for 10 min. The supernatant was collected from the samples and BCA Assay was performed to determine the protein content (Pierce™ BCA Protein Assay Kit, Thermo Scientific).

### 4.9. Western Immunoblotting (WB)

Samples were thawed on ice. Total protein (20 μg) from tissue supernatant was loaded into TGX precast gels (10% Criterion^TM^, BioRad, Hercules, CA, USA) and transferred to nitrocellulose membrane (Amersham^TM^ Protran^TM^, GE Healthcare, Chicago, IL, USA). After transfer, the membrane was blocked at room temperature for 1 h in blocking buffer (5% dry milk in Tris-buffered saline with Tween 20 (TBST)). The following primary antibodies were diluted in blocking buffer (5% BSA in TBST): β-actin (Abcam, Waltham, MA, USA, ab6276, 1:2000), FAAH (Abcam, Waltham, MA, USA, ab54615, 1:1000), AHA1 (Abcam, Waltham, MA, USA, ab83036, 1:1000), heat shock protein 90 (Cell Signaling, Danvers, MA, USA, 4874S, 1:5000, pan-antibody), heat shock protein 70 (Cell Signaling, Danvers, MA, USA, 48725, 1:7000), cannabinoid receptor 1 (Abcam, Waltham, MA, USA, ab259323, 1:500), and NAPE-PLD (ThermoScientific, Waltham, MA, USA, PA5-115616, 1:500) α-tubulin (Cell Signaling, Danvers, MA, USA, 3873S, 1:10,000). The membrane was incubated in diluted primary antibody for 48 h at 4˚C. The membrane was washed three times in TBST for 5 min each, and then incubated with Goat anti-Mouse IRDye-680 (LiCor, Lincoln, NE, USA, 926–68020) or Goat anti-Rabbit IRDye-800 (LiCor, 926-32211) in 5% milk in TBST for 1 h rocking at room temperature (25–27 °C). The membrane was washed again three times for 5 min each and imaged with an Azure Sapphire laser imager (Azure Biosystems, Dublin, OH, USA). The Western image was produced using Azure Capture software and analyzed using Azure Spot Western Analysis software. Bands were quantified using ImageJ FIJI software 2.9.0 open access from NIH.

### 4.10. RT-qPCR

Total RNA was extracted from the tissue using RNeasy Mini Kit from Qiagen (Germantown, MD, USA) (#74104) per the manufacturer’s protocol, with final reconstitution of the RNA in nuclease-free water and storage at −80 °C until analysis. The RNA was quantified using a NanoDrop ND-1000 Spectrophotometer (ThermoFisher). Single-stranded complementary DNA (cDNA) was synthesized using a high-capacity cDNA reverse transcription kit (27-874-06) from Fisher Scientific, following the manufacturer’s protocol. The amplified cDNA was added to the RT^2^ SYBR Green Master Mix (330500; Qiagen) and gene-specific primers. RT^2^ qPCR primer assays (330001; Qiagen) detected the expression of *Cnr1*, a CB1 gene, (PPR52793A), and *Cnr2*, a CB2 gene (PPR50293A), as well as *TUBA1a*, a α-tubulin gene (PPR50840B). Gene expression was normalized to α-tubulin gene expression. The quantitative PCR (qPCR) was performed on the LightCycler 96 Real-Time PCR System (Roche, Indianapolis, IN, USA), using the following thermal program: 95 °C for 10 min, 45 cycles of 95 °C for 15 s, and 60 °C for 1 min and a melt curve analysis. The melt curves were analyzed to confirm selectivity and supported by agarose gel electrophoresis.

### 4.11. ^35^S-GTPγS Coupling

The ^35^S-GTPγS coupling assay was performed as described in our previous work [50,51]. Briefly, whole PAG was Dounce-homogenized in homogenization buffer (20 mM HEPES, pH 7; 100 mM NaCl; 2 mM MgCl_2_; 1 mM EDTA; and 1 mM DTT). Tissue membrane was then centrifuged at 20,000× *g* for 30 min at 4 °C. Pellet was resuspended in 1 mL of assay buffer (20 mM HEPES, pH 7; 150 mM NaCl; 2 mM MgCl_2_; and 100 µM GDP) and Dounce-homogenized once more. Protein content was quantified by a BCA assay. 10 µg of PAG tissue membrane was combined with 25 pM of ^35^S-GTPγS (PerkinElmer, Shelton, CT, USA, #NEG030H250UC) and drug in a 200 µL reaction volume in the presence of 100 µM GDP in assay buffer. All drug dilutions and GTPγS were made in an assay buffer. The reactions were incubated at 30 °C for 2 h; then, they were collected onto GF/B filter plates (PerkinElmer) using a Brandel Cell Harvester. The plates were dried, Microscint PS (PerkinElmer) added, and the plates read in a 96-well format Microbeta2 scintillation counter (PerkinElmer). Each group was normalized to its own average baseline, and then fit to a 3-variable (Hill Slope = 1) agonist model using GraphPad Prism 9.0. The potency (EC_50_) and efficacy (E_MAX_) with the associated 95% confidence intervals were quantified and reported for each group.

### 4.12. Quantification of 2-AG and AEA by LC-MS

PAG samples for liquid chromatography/mass spectrometry (LC-MS) were purified by organic solvent extraction, as described in our previous work [44]. Briefly, tissues were harvested, snap-frozen in pre-weighted tubes, and stored at −80 °C. On the day of the experiment, samples were weighed and homogenized in 1 mL of chloroform/methanol (2:1 *v*/*v*) supplemented with phenylmethylsulfonyl fluoride (PMSF) at 1 mM final concentration to inhibit the degradation of endocannabinoids by endogenous enzymes. The homogenization was carried out by a Dounce homogenizer. Homogenates were then mixed with 0.3 mL of 0.7% *w*/*v* NaCl, vortexed, and then centrifuged for 10 min at 3200× *g* at 4 °C. The aqueous phase and debris were collected and extracted two more times with 0.8 mL of chloroform. The organic phases from the three extractions were pooled together and AN internal standard was added to each sample. The mixture of internal standard solutions were prepared by serial dilution of d^4^-AEA- and d^5^-2-AG- in acetonitrile. Furthermore, 6 µL of 30% glycerol in methanol per sample was added to each sample, and then the organic solvents were evaporated under nitrogen gas. Dried samples were reconstituted with chloroform (0.2 mL) and mixed with 1 mL of ice-cold acetone to precipitate proteins. The mixtures were then centrifuged for 5 min at 1800× *g* at 4 °C. The organic layer of each sample was collected and evaporated under nitrogen. Analysis of 2-AG and AEA was performed on an Ultivo triple quadrupole mass spectrometer combined with a 1290 Infinity II UPLC system (Agilent, Palo Alto, CA, USA), as described in our previous paper [44]. The instrument was run in electrospray positive mode with a gas temperature of 150 °C at a flow of 5 L/min, nebulizer at 15 psi, capillary voltage of 4500 V, sheath gas at 400 °C with a flow of 12 L/min and a nozzle voltage of 300 V. The following transitions were monitored: 348.3→287.3 and 62, 352.3→287.4 and 65.9, 379.3→287.2 and 269.2, and 384.3→287.2 and 296.1 for AEA, d^4^-AEA, 2-AG, and d^5^-2-AG-. The first fragment listed was utilized for quantification and the second fragment was applied for confirmation. The first 3 min of analysis time was diverted to waste. Chromatographic separation used the following isocratic system: 21% 1 mM ammonium fluoride and 79% methanol on an Acquity UPLC BEH C-18 1.7u 2.1 × 100 mm column (Waters, Milford, MA, USA) maintained at 60 °C. After each injection, the column was washed with 90% methanol for one minute and then re-equilibrated for 5 min prior to the next injection. Samples were maintained at 4 °C. Mixed calibration solutions were made by serial dilution of AEA and 2-AG stock solutions in 80% C_2_H_3_N. Calibration curves for each analysis were prepared by mixing 10 µL of internal standard solution and 20 µL of standard solution. Prior to analysis, dried samples were dissolved in 200 µL of 80:20 C_2_H_3_N:H_2_O then vortexed and sonicated, followed by centrifugation at 15,800× *g* at 4 °C for 5 min. Supernatant was transferred to autosampler vials and 5 µL was injected for analysis.

### 4.13. Statistical Analysis

GraphPad Prism 9.5.0 software (GraphPad Software) was used for statistical analysis. To determine the numbers needed for each experiment, G.Power3.1 was used for 80% power to detect a 20% effect (ρ = 0.2) when alpha = 0.05. Unless otherwise stated, the data were expressed as mean ± S.E.M. Groups were assessed for normality and sphericity before statistical comparison. Periorbital allodynia measurements were assessed using a repeated-measure two-way ANOVA to analyze differences between treatment groups over time with a Bonferroni test applied post hoc [27]. Molecular studies were compared by unpaired *t*-test or one-way ANOVA with Tukey’s post-test, as indicated. Differences were considered significant if *p* ≤ 0.05. To limit a biased interpretation of data, researchers conducting and analyzing the experiments were blinded to the protocol, with another researcher responsible for randomizing and treating rats to each experiment (e.g., double-blind).

## 5. Conclusions

Before this work, an interaction between Hsp90 and the AEA system was not fully recognized. Our results confirmed the anti-allodynic effect of the Hsp90 inhibitor 17-AAG in a preclinical model of migraine by modulation of the FAAH-NAPE-PLD-AEA system in PAG. The more complete understanding of the connection between Hsp90 and the endocannabinoid system can provide guidance to find new drug approaches in targeting migraine pain. Our data also highlight the importance of the endocannabinoid system in migraine pain, in agreement with former research.

## Figures and Tables

**Figure 1 pharmaceuticals-18-01430-f001:**
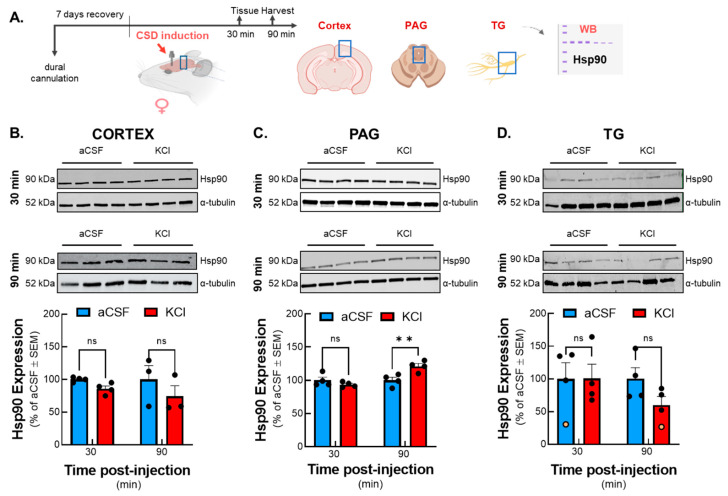
**Total expression of Hsp90 was elevated in PAG harvested at 90 min after CSD induced by cortical KCl injection.** Dural cannulation was performed on female Sprague Dawley rats. After 7 days of recovery, focal injection of 0.5 µL of 1 M KCl or artificial CSF (aCSF) was delivered through the dural cannula into the cerebral cortex. Brain tissue was harvested from regions of the brain associated with CSD pain (cortex, PAG, and TG) at two different time points (30 and 90 min) post-aCSF or KCl injections, then Western immunoblotting was performed to measure the expression of Hsp90. (**A**) Schematic of measuring the expression of Hsp90 in cortex, PAG, and TG samples after CSD induction. Representative images of cortex (**B**), PAG (**C**), and TG (**D**) samples showing the relative expression of Hsp90 and α-tubulin as loading control at 30 and 90 min time points after cortical injections. Quantification of Western blot images revealed no significant differences between KCl and aCSF-treated tissue in cortex and TG at any time points. Elevated expression of Hsp90 was observed in PAG samples 90 min after KCl injection compared to aCSF control. All data represent the % of aCSF relative expression ± SEM (n = 3–4 in each group). ** denotes significantly different (*p* < 0.01) aCSF vs. KCl, as assessed by two-way ANOVA. ns = non-significant. Dots indicate individual biological replicates; orange denotes statistically identified outlier, ROUT 10%.

**Figure 2 pharmaceuticals-18-01430-f002:**
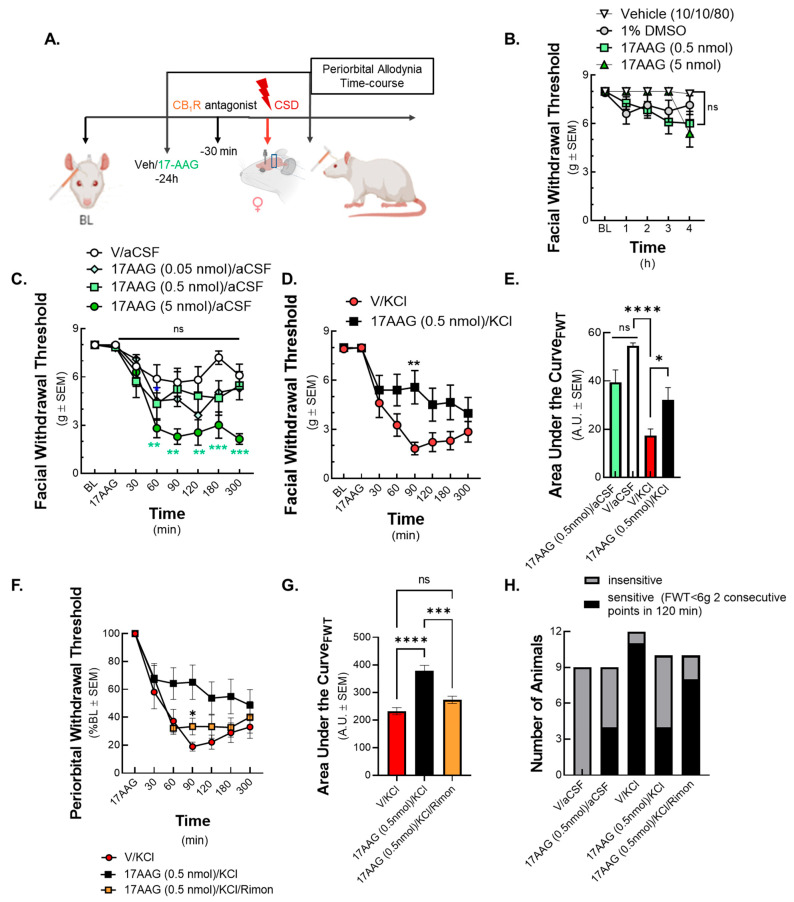
**Hsp90 inhibition with 17-AAG applied over dura mater alleviates periorbital allodynia caused by CSD.** Female Sprague Dawley rats were utilized for behavior assays 7 days after implantation of dural canula. 17-AAG or vehicle was injected through the guide canula over dura mater 24 h before CSD induction. CSD was induced by cortical injection of KCl or aCSF control. Facial sensitivity was measured by von Frey filaments at baseline, after injection of 17-AAG (acute response), and different time points after KCl injection. (**A**) Panel shows the schematic representation of the experimental design. (**B**) The Hsp90 inhibitor 17-AAG at two different doses (0.5 nmol and 5 nmol) or vehicle (1% DMSO in saline or DMSO/Tween 20/saline (1:1:8)) was injected through the guide canula over dura mater. Periorbital allodynia was assessed by von Frey filaments at baseline, 1, 2, 3, and 4 h after injection. No significant changes in facial withdrawal threshold were observed between any doses of 17-AAG vs. vehicle groups as assessed by two-way ANOVA, concluding that acute administration of 17-AAG did not induce periorbital allodynia. Values are mean ± SEM (n = 8–12 in each group). (**C**) In the next set of experiment, 17-AAG at three different doses (0.05 nmol, 0.5 nmol, and 5 nmol) or vehicle was injected through the canula, followed by cortical aCSF (0.5 μL) injection via dural canula 24 h later. Facial sensitivity was measured by the von Frey test at baseline, after injection of 17-AAG (acute response), and 30, 60, 90, 120, 180, and 300 min after aCSF administration. 17-AAG at the highest dose (5 nmol) in combination with aCSF induced periorbital allodynia at 60, 90, 120, 180, and 300 min time points compared to vehicle + aCSF control. However, Hsp90 inhibitor at lower doses (0.05 nmol and 0.5 nmol) did not significantly influence facial withdrawal thresholds. Values are mean ± SEM (n = 8–12 in each group). ** *p* < 0.01, *** *p* < 0.001, compared to vehicle + aCSF control as assessed by two-way ANOVA. (**D**) In the next set of animals, 17-AAG at 0.5 nmol or vehicle was applied on the dura 24 h before CSD induction. CSD was induced by cortical injection of KCl (1M). Facial allodynia was assessed by von Frey test at baseline, after injection of 17-AAG, and 30, 60, 90, 120, 180, and 300 min after cortical KCl injection. 17-AAG (0.5 nmol) significantly increased facial withdrawal threshold at 90 min time point, alleviating CSD-caused allodynia. Values are mean ± SEM (n = 11–12 in each group). ** *p* < 0.01 compared to KCl as assessed by two-way ANOVA. (**E**) Area under the curve of 17-AAG (0.5 nmol) + aCSF, vehicle + aCSF, vehicle + KCl, and 17-AAG (0.5 nmol) + KCl group obtained from the corresponding behavior assays. * *p* < 0.05, **** *p* < 0.0001 as assessed by one-way ANOVA. (**F**) In this experimental setting, 17-AAG (0.5 nmol) or vehicle was injected through dural canula 24 h before KCl injection. 30 min before CSD induction, the CB_1_R antagonist rimonabant (1 mg/kg) was administered intraperitoneally. Facial allodynia was measured by von Frey test at baseline, after injection of 17-AAG, and 30, 60, 90, 120, 180, and 300 min after cortical KCl injection. Rimonabant significantly decreased the facial withdrawal threshold at 90 min compared to 17-AAG/KCl group, suggesting the CB_1_ receptor dependency of the anti-allodynic effect of Hsp90 inhibition. Values are mean ± SEM (n = 9–12 in each group). * *p* < 0.05 17/KCl vs. 17/KCl/rimonabant group as assessed by two-way ANOVA. (**G**) Area under the curve comparing all groups. *** *p* < 0.001, **** *p* < 0.0001, ns = non-significant as assed by one-way ANOVA. (**H**) The graph compares the number of animals showing or not showing facial sensitivity. The facial sensitivity was defined FWT < 6 g at two consecutive time points after cortical injection. The exact number of animals used in behavior experiments are shown in Appendix A.

**Figure 3 pharmaceuticals-18-01430-f003:**
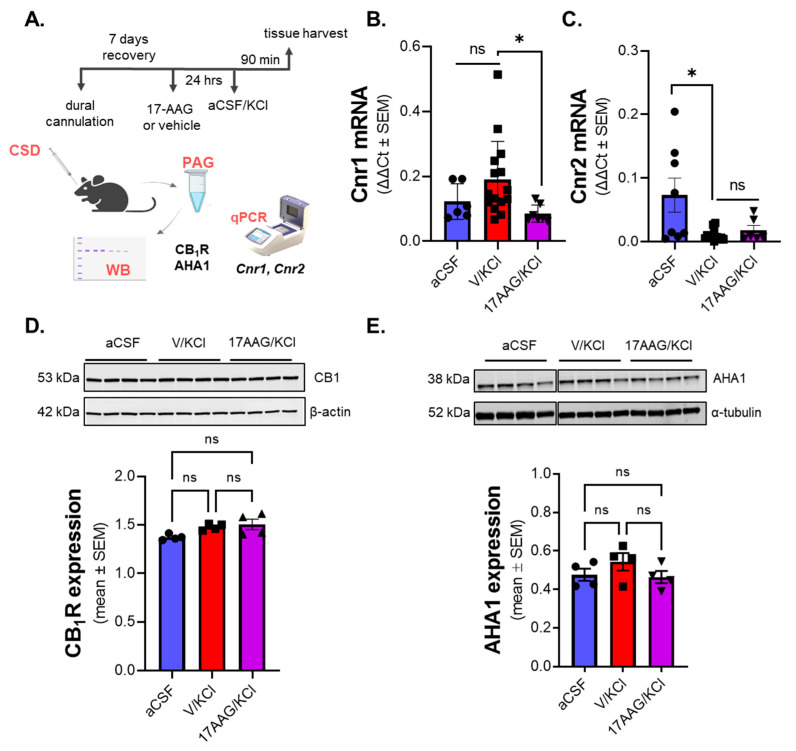
**Expression of cannabinoid receptors and AHA1 in PAG after Hsp90 inhibition in the CSD model.** Female Sprague Dawley rats were implanted with dural cannula. 17-AAG (0.5 nmol) or vehicle (1% DMSO in saline) were applied to the dura mater 24 h before cortical KCl injection. Then, 90 min after cortical injection of KCl or aCSF, PAG tissue was harvested and then subjected to either qPCR or Western immunoblotting. (**A**) Schematic of experimental process to detect changes in the expression of cannabinoid receptors and AHA1 in the CSD model. (**B**) No significant difference in the mRNA level of the CB_1_R (*Cnr1* gene) was observed between the aCSF and KCl groups. However, there was a significant decrease in the *Cnr1* mRNA in 17-AAG pretreated KCl group compared to the KCl alone as assessed by one-way ANOVA. Values are mean ± SEM, relative to the expression of *Tuba1a* gene (n = 6–16/group). * Denotes significantly different (*p* < 0.05) KCl + vehicle vs. 17-AAG + KCl, as assessed by one-way ANOVA. ns = non-significant. (**C**) The pretreatment of 17-AAG did not cause significant changes in the mRNA level of the CB_2_R (*Cnr2* gene) as compared to the KCl group. A significant decrease in *Cnr2* mRNA was observed between aCSF vs. KCl as assessed by one-way ANOVA. Values are mean ± SEM, relative to the expression of the *Tuba1a* gene (n = 6–12/group). * Denotes significantly different (*p* < 0.05) aCSF vs. KCl, as assessed by one-way ANOVA. ns = non-significant. (**D**) Representative images showing protein levels of CB_1_R and β-actin as loading control in PAG samples. There were no significant differences among groups, as assessed by one-way ANOVA. ns = non-significant. Values are mean ± SEM, relative to the expression of β-actin (n = 4/group). (**E**) The expression of AHA1 was also assessed by Western blot. Images show the expression of AHA1 and α-tubulin as loading control in PAG samples. No significant differences in the expression of AHA1 were detected among groups, as assessed by one-way ANOVA. ns = non-significant. Values are mean ± SEM, relative to the expression of α-tubulin (n = 4 in each group).

**Figure 4 pharmaceuticals-18-01430-f004:**
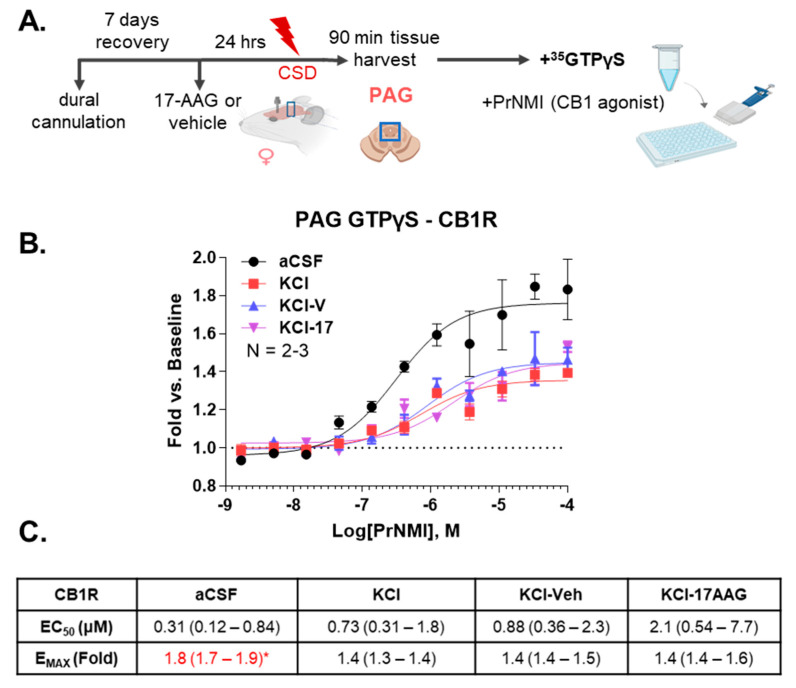
**^35^S-GTPγS coupling assay using PrNMI to test CB_1_R functionality in PAG tissue harvested 90 min after cortical insult.** aCSF or KCl (1M) were administered to female Sprague Dawley rats via dural canula. 17-AAG (0.5 nmol) or vehicle (1% DMSO in saline) were injected onto the dura mater 24 h before KCl injection. Then, 90 min after cortical injection of KCl or aCSF, PAG tissue was collected and then subjected to the ^35^S-GTPγS coupling assay. (**A**) Schematic outline of experimental design. (**B**) The CB_1_R selective compound PrNMI was used in agonist mode up to 100 μM. Data represented as the mean ± SEM of N = 2–3 animals/group. (**C**) The potency (EC_50_) and efficacy (E_MAX_) were calculated from each summary curve consisting of N = 2–3 independent animals per group, reported as the calculated value with 95% confidence intervals in parentheses. A significant difference as indicated by non-overlapping confidence intervals with other group values was noted by red highlight and *. These findings show a loss of efficacy in the CB_1_R system with the KCl treatment that is not rescued by 17-AAG administration.

**Figure 5 pharmaceuticals-18-01430-f005:**
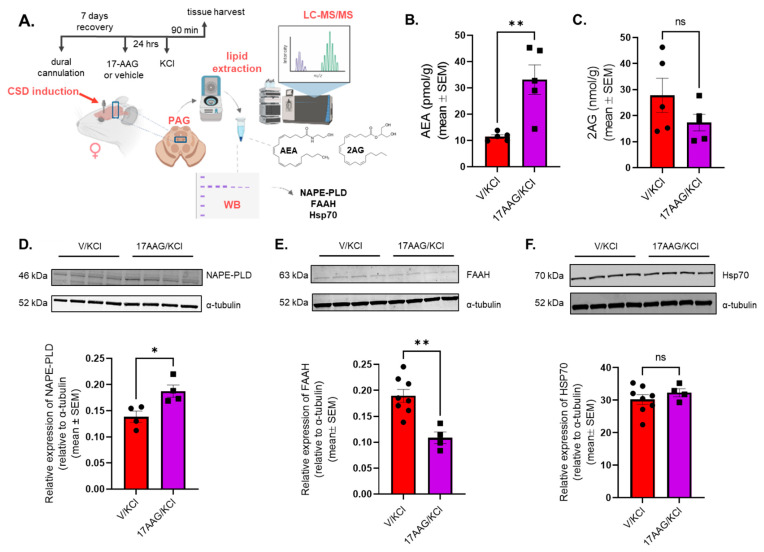
**Hsp90 inhibition increased the level of AEA and coordinately regulated NAPE-PLD and FAAH expression in PAG after CSD induction.** Female Sprague Dawley rats had dural cannulas implanted. After 7 days of recovery, 5 µL of 17-AAG (0.5 nmol) or vehicle (1% DMSO in saline) was injected on the dura mater. After 24 h, KCl (1M) was administered cortically via dural canula to all animals. PAG samples were harvested 90 min after KCl injection. The samples were subjected to lipid extraction for LC-MS to measure endocannabinoids. A separate set of animals were used for Western blot to detect NAPE-PLD, FAAH, and Hsp70 expression. (**A**) Panel shows the outline of the experimental protocol. (**B**) A significant increase in AEA levels was detected in PAG samples treated with 17-AAG + KCl compared to vehicle + KCl control, as assessed by unpaired *t*-test, ** *p* < 0.01. Values are mean ± SEM (n = 5/condition). (**C**) 2-AG levels were not significantly different between vehicle + KCl vs. 17-AAG + KCl groups as assessed by unpaired *t*-test. ns= non-significant. Values are mean ± SEM (n = 5/condition). (**D**) Representative images of Western immunoblotting detecting NAPE-PLD and α-tubulin as loading control in PAG samples. A significant increase in NAPE-PLD expression was detected in PAG samples of 17-AAG + KCl group compared to vehicle + KCl, as assessed by unpaired *t*-test, * *p* < 0.05. Values are mean ± SEM, relative to the expression of α-tubulin (n = 4/group). (**E**) Representative images showing the expression of FAAH and α-tubulin as loading control in PAG samples. The pretreatment of 17-AAG caused a significant reduction in FAAH expression in PAG samples compared to vehicle + KCl, as assessed by unpaired *t*-test, ** *p* < 0.01. Values are mean ± SEM, relative to the expression of α-tubulin (n = 4–8/group). (**F**) Representative images of Western immunoblotting detecting Hsp70 and α-tubulin as loading control in PAG samples. No significant difference was observed between vehicle + KCl vs. 17-AAG + KCl, as assessed by unpaired *t*-test. ns = non-significant. Values are mean ± SEM, relative to the expression of α-tubulin (n = 4–8/group).

## Data Availability

Data presented in this study is contained within the article and Appendix A. Further inquiries can be directed to the corresponding author.

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
