# Peer review of "Brain Hsp90 Inhibition Mitigates Facial Allodynia in a Rat Model of CSD Headache and Upregulates Endocannabinoid Signaling in the PAG"

_pharmaceuticals, 2025, doi:10.3390/ph18101430_

Round 1

Reviewer 1 Report

Comments and Suggestions for Authors

The reviewer would like to declare no conflict of interest with the authors and their affiliations.

Generally, the manuscript is well-written, and the results are well presented. However, some minor technical concerns need to be addressed before the manuscript can be recommended for acceptance.

  1. For each figure, its content should be self-explanatory by its figure caption. In the manuscript, the figure captions appear to be 1-line descriptions. However, it is noted that a full paragraph underneath each figure caption is present, but it may create confusion to the readers on where to find the corresponding figure captions.
  2. For the von frey measurement, the author did not specify whether the rats were free-moving or being restrained when the "von Frey filaments perpendicularly positioned to the midline of the forehead".  Also, please specify how many readings were captured at each meansurement timepoint, 1? 3? ad how was the results recorded as? mean of 3 readings? also, it is important to specify the time interval between each reading.
  3. in the "tissue harvest" section, the author mentioned that "PAG was dissected". It should be specified whether the whole PAG was collected? or the brain being sliced, and specific region of the PAG tissue was collected?
  4. Given that the whole PAG can be anatomically diverted into different sections, which each govern different (although some overlapping) physiological functions. the authors should mention which part of the PAG is being studied. If whole PAG was being used, justification should be provided.
  5. For western blotting, LC/MS and PCR studies, how much of the PAG tissue was being used? The signaling substance were extracted from how much volume of PAG tissue?
  6. For better understanding of the proposed interaction of Hsp90 and cannabinoid signaling in the PAG for migraine modulation, it is recommended to include a figure/diagramme showing the proposed signaling pathway.
  7. The baseline level of expression of AEA, 2-AG, FAAH, NAPE-PLD and Hsp90, as well as allodynic response of non-CSD induced mice are missing. These baseline data are important to further validate the participation of the Hsp90 and/or cannabinoid signaling in this study model  

Author Response

Response to Reviewer 1

The authors would like to thank the senior editor and the reviewers for their constructive comments and suggestions. The authors’ answers for each comment (in Italic, highlighted with grey) can be found below. The changes and edits in the manuscript are highlighted in yellow.

  1. For each figure, its content should be self-explanatory by its figure caption. In the manuscript, the figure captions appear to be 1-line descriptions. However, it is noted that a full paragraph underneath each figure caption is present, but it may create confusion to the readers on where to find the corresponding figure captions.

The 1-line description is supposed to be the title of each figure and the full paragraph is supposed to serve as a caption. We have made some modifications in the revised manuscript, so it is more obvious that they belong together.

  1. For the von frey measurement, the author did not specify whether the rats were free-moving or being restrained when the "von Frey filaments perpendicularly positioned to the midline of the forehead".  Also, please specify how many readings were captured at each meansurement timepoint, 1? 3? ad how was the results recorded as? mean of 3 readings? also, it is important to specify the time interval between each reading.

The method section of “Assessment of periorbital mechanical allodynia” has been modified based on the reviewer`s suggestions.  

  1. in the "tissue harvest" section, the author mentioned that "PAG was dissected". It should be specified whether the whole PAG was collected? or the brain being sliced, and specific region of the PAG tissue was collected?

The coordinates of PAG collection were added to the method section.

  1. Given that the whole PAG can be anatomically diverted into different sections, which each govern different (although some overlapping) physiological functions. the authors should mention which part of the PAG is being studied. If whole PAG was being used, justification should be provided.

The coordinates of PAG collection were added to the method section.

  1. For western blotting, LC/MS and PCR studies, how much of the PAG tissue was being used? The signaling substance were extracted from how much volume of PAG tissue?

The entire piece of PAG samples were used to each molecular study. The weight of PAG samples were recorded for LC-MS experiment with average weight of 30-35 mg per sample.

  1. For better understanding of the proposed interaction of Hsp90 and cannabinoid signaling in the PAG for migraine modulation, it is recommended to include a figure/diagramme showing the proposed signaling pathway.

A graphical abstract was added to the manuscript which shows the proposed signaling pathway.

  1. The baseline level of expression of AEA, 2-AG, FAAH, NAPE-PLD and Hsp90, as well as allodynic response of non-CSD induced mice are missing. These baseline data are important to further validate the participation of the Hsp90 and/or cannabinoid signaling in this study model.

While we appreciate the reviewers comment, all components of the ECS in naive rats have been published (https://pubmed.ncbi.nlm.nih.gov/34749819), along with changes of ECS in CSD model (https://pubmed.ncbi.nlm.nih.gov/37287623/). The appropriate controls for this study were carefully considered with aCSF injection being determined as appropriate. This was determined as aCSF does not induce CSD (https://pubmed.ncbi.nlm.nih.gov/30073201/) yet accounts for dural damage that may influence Hsp90 and the ECS.

Reviewer 2 Report

Comments and Suggestions for Authors

The manuscript is of high quality and presents a cohesive, impactful, and quite well-supported scientific narrative. The presented findings are novel and of clear interest to a broad audience of researchers in the field of neurosciences and pharmacology. 

However, several points related to the conceptual interpretation of the 24-hour pre-treatment paradigm and some minor issues in data presentation and discussion warrant clarification and revision. 

  1. Material and Methods

4.2. Animals

For clarity, it would be useful to include a description and a table showing the experimental setup, with precise grouping of animals (not just the total number “n”). 

4.11. 35S-GTPγS Coupling

What was the composition of the assay buffer? Was it a commercial buffer or a self-made one? 

4.5. Pre-cortical injection treatments

High-dose 17-AAG (5 nmol) uses DMSO/Tween/saline, whereas lower doses use saline+DMSO. Ensure that each dose is compared to its matched vehicle and avoid cross-vehicle contrasts. A single sentence explaining this difference would be useful for the correct interpretation of this section and, consequently, of the results discussed further in this context.

WB normalization and interpretation

For Westerns, show full uncropped blots with MW markers (you provided originals) and map all lanes used for quantification - uncropped files show excluded lanes (X). Please annotate reasons in the main figure or legend and include MW ladders. Ensure that housekeeping bands correspond lane-for-lane; justify excluded lanes (marked “X” in the provided raw blots) and align them with the plotted numbers. Consider using dot plots with individual values (proposal for consideration only). 

Discussion

I think the Discussion should be somehow expanded to elaborate on the potential functional significance of this PAG-specific Hsp90 upregulation. Perhaps a few sentences of justification would be useful here, to emphasize more significantly the selection of PAG as a primary site of central sensitization in the presented CSD model and Hsp90 as a key molecular mediator of this process - this would add a compelling layer of conceptual depth to their findings and their implications for understanding headache pathophysiology.

The Significance of the 24-Hour Pre-treatment 

A central feature of the experimental design is the administration of the Hsp90 inhibitor 17-AAG, which is given 24 hours prior to the induction of CSD. This methodological choice has profound implications for the interpretation of the study's core mechanistic findings -> Hsp90's canonical function as a molecular chaperone involves the post-translational stabilization and regulation of client protein conformation and activity, processes that, I think, typically occur on a timescale of minutes - hours. However, a 24-hour pre-treatment window appears to be an ideal timeframe for observing slower, adaptive cellular processes, such as changes in gene transcription and subsequent protein translation and turnover.   

The results of the study align perfectly with a mechanism of altered protein expression rather than acute signaling modulation. The key molecular changes observed following 17-AAG treatment were alterations in the total protein levels of the metabolic enzymes NAPE-PLD (increased) and FAAH (decreased). Conversely, the study found no evidence that 17-AAG acutely modulated the intrinsic signaling capacity of the CB1 receptor itself - the GTPγS functional assay showed that, while CSD desensitized the receptor, 17-AAG did not reverse this effect. Such observations suggest that the experimental design itself predisposed the results toward a mechanism of altered protein expression. The 24-hour pre-treatment likely allowed 17-AAG to modulate the activity of Hsp90-related proteins that are themselves regulators of gene expression, such as transcription factors. The authors briefly allude to this possibility in the Discussion, mentioning Hsp90's known role in regulating transcription factors like HNF4A (a promoter of NAPE-PLD expression) and STAT3 (a promoter of FAAH expression). I believe this connection is crucial and should be highlighted in the discussion to more explicitly address the importance of the 24-hour pre-treatment interval. The authors should briefly highlight how this timeframe strongly supports a mechanism of transcriptional and/or translational regulation of AEA metabolic enzymes, as opposed to an acute modulation of signaling protein function. Highlighting this point would not only strengthen the interpretation of their data but also showcase the elegance and foresight of their experimental design.

Author Response

Response to Reviewer 2

The authors would like to thank the senior editor and the reviewers for their constructive comments and suggestions. The authors’ answers for each comment (in Italic, highlighted with grey) can be found below. The changes and edits in the manuscript are highlighted in yellow.

  1. Material and Methods

4.2. Animals

For clarity, it would be useful to include a description and a table showing the experimental setup, with precise grouping of animals (not just the total number “n”). 

A supplemental table with the number of animals in each group of behavior assay was added. The number of animals in the molecular work is shown in the individual graphs.

4.11. 35S-GTPγS Coupling

What was the composition of the assay buffer? Was it a commercial buffer or a self-made one? 

The composition of assay buffer was added to the revised method section.

4.5. Pre-cortical injection treatments

High-dose 17-AAG (5 nmol) uses DMSO/Tween/saline, whereas lower doses use saline+DMSO. Ensure that each dose is compared to its matched vehicle and avoid cross-vehicle contrasts. A single sentence explaining this difference would be useful for the correct interpretation of this section and, consequently, of the results discussed further in this context.

We encountered solubility difficulties with the highest dose of 17-AAG, thus different vehicle composition was used for that dose. The possible effect of different vehicles on behavior assay was tested, no significant differences were found (see in Fig 2B).

WB normalization and interpretation

For Westerns, show full uncropped blots with MW markers (you provided originals) and map all lanes used for quantification - uncropped files show excluded lanes (X). Please annotate reasons in the main figure or legend and include MW ladders. Ensure that housekeeping bands correspond lane-for-lane; justify excluded lanes (marked “X” in the provided raw blots) and align them with the plotted numbers. Consider using dot plots with individual values (proposal for consideration only). 

The uncropped blots are provided in the supplemental file and further notes were added. The excluded lanes in the uncropped images represent samples harvested from animals with different treatment that is not relevant for this study. Those bands were not part of this study therefore they were not quantified and not added to the graphs. All the lanes included in this study are numbered in the uncropped images and notes about which lanes are shown in the main figures were added to the uncropped images (all lanes except for Fig5E and 5F are shown in the main figures). MW ladder is always shown in the uncropped images, moreover the expected band sizes are indicated in the main figures. The individual values are shown in each graph for molecular work.

Discussion

I think the Discussion should be somehow expanded to elaborate on the potential functional significance of this PAG-specific Hsp90 upregulation. Perhaps a few sentences of justification would be useful here, to emphasize more significantly the selection of PAG as a primary site of central sensitization in the presented CSD model and Hsp90 as a key molecular mediator of this process - this would add a compelling layer of conceptual depth to their findings and their implications for understanding headache pathophysiology.

The Discussion section in the revised manuscript has been expanded using the comments above.

The Significance of the 24-Hour Pre-treatment 

A central feature of the experimental design is the administration of the Hsp90 inhibitor 17-AAG, which is given 24 hours prior to the induction of CSD. This methodological choice has profound implications for the interpretation of the study's core mechanistic findings -> Hsp90's canonical function as a molecular chaperone involves the post-translational stabilization and regulation of client protein conformation and activity, processes that, I think, typically occur on a timescale of minutes - hours. However, a 24-hour pre-treatment window appears to be an ideal timeframe for observing slower, adaptive cellular processes, such as changes in gene transcription and subsequent protein translation and turnover.   

The results of the study align perfectly with a mechanism of altered protein expression rather than acute signaling modulation. The key molecular changes observed following 17-AAG treatment were alterations in the total protein levels of the metabolic enzymes NAPE-PLD (increased) and FAAH (decreased). Conversely, the study found no evidence that 17-AAG acutely modulated the intrinsic signaling capacity of the CB1 receptor itself - the GTPγS functional assay showed that, while CSD desensitized the receptor, 17-AAG did not reverse this effect. Such observations suggest that the experimental design itself predisposed the results toward a mechanism of altered protein expression. The 24-hour pre-treatment likely allowed 17-AAG to modulate the activity of Hsp90-related proteins that are themselves regulators of gene expression, such as transcription factors. The authors briefly allude to this possibility in the Discussion, mentioning Hsp90's known role in regulating transcription factors like HNF4A (a promoter of NAPE-PLD expression) and STAT3 (a promoter of FAAH expression). I believe this connection is crucial and should be highlighted in the discussion to more explicitly address the importance of the 24-hour pre-treatment interval. The authors should briefly highlight how this timeframe strongly supports a mechanism of transcriptional and/or translational regulation of AEA metabolic enzymes, as opposed to an acute modulation of signaling protein function. Highlighting this point would not only strengthen the interpretation of their data but also showcase the elegance and foresight of their experimental design.

The Discussion section in the revised manuscript has been expanded highlighting the importance of the 24-hour pretreatment.

Round 2

Reviewer 1 Report

Comments and Suggestions for Authors

The authors have addressed my previous comments.